# Transcriptome-Based WGCNA Analysis Reveals Regulated Metabolite Fluxes between Floral Color and Scent in *Narcissus tazetta* Flower

**DOI:** 10.3390/ijms22158249

**Published:** 2021-07-31

**Authors:** Jingwen Yang, Yujun Ren, Deyu Zhang, Xuewei Chen, Jiazhi Huang, Yun Xu, Cristina Belén Aucapiña, Yu Zhang, Ying Miao

**Affiliations:** Fujian Provincial Key Laboratory of Plant Functional Biology, College of Life Sciences, Fujian Agriculture and Forestry University, Fuzhou 350002, China; 2160539002@fafu.edu.cn (J.Y.); ryj@fafu.edu.cn (Y.R.); zhangdeyu@fafu.edu.cn (D.Z.); 1180514008@fafu.edu.cn (X.C.); 3185403009@stu.fafu.edu.cn (J.H.); 1190514066@fafu.edu.cn (Y.X.); crisbel.ca09@gmail.com (C.B.A.)

**Keywords:** transcriptome-based WGCNA analysis, metabolite flux, floral pigment, floral fragrance, transcription factors, *Narcissus tazetta*

## Abstract

A link between the scent and color of *Narcissus tazetta* flowers can be anticipated due to their biochemical origin, as well as their similar biological role. Despite the obvious aesthetic and ecological significance of these colorful and fragrant components of the flowers and the molecular profiles of their pigments, fragrant formation has addressed in some cases. However, the regulatory mechanism of the correlation of fragrant components and color patterns is less clear. We simultaneously used one way to address how floral color and fragrant formation in different tissues are generated during the development of an individual plant by transcriptome-based weighted gene co-expression network analysis (WGCNA). A spatiotemporal pattern variation of flavonols/carotenoids/chlorophyll pigmentation and benzenoid/phenylpropanoid/ monoterpene fragrant components between the tepal and corona in the flower tissues of *Narcissus tazetta*, was exhibited. Several candidate transcription factors: MYB12, MYB1, AP2-ERF, bZIP, NAC, MYB, C2C2, C2H2 and GRAS are shown to be associated with metabolite flux, the phenylpropanoid pathway to the production of flavonols/anthocyanin, as well as related to one branch of the phenylpropanoid pathway to the benzenoid/phenylpropanoid component in the tepal and the metabolite flux between the monoterpene and carotenoids biosynthesis pathway in coronas. It indicates that potential competition exists between floral pigment and floral fragrance during *Narcissus tazetta* individual plant development and evolutionary development.

## 1. Introduction

The cultivated variety “Jinzhanyintai” of *Narcissus tazetta* var. *chinensis* belongs to the Amaryllidaceous family and is a perennial bulbous plant widely cultivated in East Asia and China, and is one of the ten most famous traditional flowers in China, with its beautiful flower and rich fragrance serving high ornamental value [1,2].

Flower color and scent are not only commercially important, in which they influence the yield and quality of many crops, but they are also important in their commercial appeal. In nature, the color and fragrance of flowers are two of the main characteristics adopted for plants to attract pollinators to ensure the reproductive success of the plant [3,4,5,6,7]. Floral scent is a combination characteristic determined by a complex mixture of volatile molecules, which can be classified by their biosynthetic origin into terpenes, phenylpropanoids and fatty acid derivatives [8,9]. Numerous structural genes responsible for the formation of volatile compounds have been identified [10,11], including genes responsible for the formation of phenylpropanoid and benzenoid compounds [12,13,14,15] and for the formation of monoterpenes and carotenoids/chlorophyll compounds [16,17,18,19,20]. To date, knowledge about the regulation of floral fragrance biosynthesis is limited, and only a few transcription factors, ODORANT1, EOBII, and PhMYB4, have been shown to be involved in the regulation of volatile biosynthesis in flowers [10,11,21,22,23]. Another group of metabolites determining feature traits are flavonol/anthocyanin pigments, which derive from a well-defined branch of the phenylpropanoid pathway. Numerous structural and regulatory genes involved in anthocyanin biosynthesis have been extensively used for the genetic manipulation of floral color [24,25]. The regulation of anthocyanin biosynthesis has been shown to occur primarily through the action of MYB transcription factors [26]. However, the activities of these regulators are not restricted to the regulation of anthocyanin production, but also have an effect on fragrance formation or tissue development.

A link between pigment and fragrance can be anticipated due to their biochemical origin, as well as their similar biological role. The diversion of metabolic flux from one branch of phenylpropanoid pathway to another has been reported in *Petunia* [27]. The transferring of transcription factor PAP1 from *Arabidopsis* into *Petunia* can cause the simultaneous enhancement of both branches of the phenylpropanoid pathway, leading to the production of color and scent in flowers [8]. Studies on *Petunia* have revealed a conserved mechanism for the formation of benzenoid/phenylpropanoid components and flavonol/anthocyanin pigmentation in petals. Pigments of anthocyanin are only produced in the overlapping expression domains of the the R2R3-MYB and bHLH coregulators of anthocyanin biosynthetic genes [28,29]. Further research reported that the *PhMYB12* locus encodes an R2R3-MYB transcription factor that regulates the expression of *CINNAMIC ACID 4-HYDROXYLASE* (*C4H*)*, 4-COUMARATE COA LIGASE* (*4CL*) and, *FLAVONOL SYNTHASE* (*FLS*), which presumably redirects metabolite flux from anthocyanin biosynthesis to the production of colorless flavonoids, indicating that a low level of *FLS* expression in the petal lobe and a consequent absence of the spatial patterning in *Petunia* flowers [30]. Despite this, the molecular profiles of pigment and fragrant formation have been addressed in some plants. However, the molecular basis of the correlation of fragrant components and color pattern is still limited.

In this study, we simultaneously use one way to address that the floral pigmentation and fragrance formation are generated during the development of an individual plant, and that their patterns, which are diversified among tissues, are generated from different gene expressions during development. Using this approach, we analyzed a spatiotemporal pattern variation of flavonols/carotenoids pigmentation and fragrant component between the tepal and corona tissues of *Narcissus tazetta* by transcriptome-based weighted gene co-expression network analysis (WGCNA). Several candidate transcription factors are shown to associate the metabolite flux of the phenylpropanoid pathway with the production of colorless flavonols, from one branch of the phenylpropanoid pathway to the benzenoid/phenylpropanoid component in the tepal, and the metabolite flux between monoterpene and carotenoids biosynthesis pathway in the corona. It indicates that the potential competition exists between floral pigment and floral fragrance during the individual plant development of *Narcissus tazetta* and evolutionary development.

## 2. Results

### 2.1. Floral Color and Scent Trait Profile in Narcissus Tepals and Coronas

In order to characterize the floral pigment and fragrant component of the *Narcissus tazetta* cultivated variety “Jinzhanyintai”, flower-development processes including the tepal and corona of *Narcissus* were observed during a full growing period. Flower buds sprout in bulbs after summer dormancy; the newly formed flower buds are nearly colorless. After five days of planting, bulbs grow shoots with sheathed leaves and after 20-day spathes are completely dehiscent; the color of tepals gradually fades, turning from deep-green to white [2]. The flower buds open and the tepal color in mature flowers turns to pure white and the volatile scent emits simultaneously. During this whole period (about 20 days), the floral color experiences five typical stages: for tepals, the color goes from colorless (T0), yellowish (T1), dark green (T2), and green-white transformation (T3) to the pure white (T4) stages; for coronas, it starts to separate from the tepals (C1) and then shows yellowish (C1), dark green (C2), green-yellow (C3) to yellow (C4) (Figure 1A). Meanwhile, at the C3 and T3 stages, the flowers start openning and emitting fragrance. Therefore, tepals and coronas at four stages, T1 (C1), T2 (C2), T3 (C3) and T4 (C4), were collected for subsequent fragrance and pigment metabolite analysis (Figure 1A).

Based on a previous report, floral pigments in *Narcissus* tepal and corona were carotenoids and flavonols only and anthocyanin was not present [31]. To determine the pigment componentchanging profile of *Narcissus* tepals and coronas during flower development, we used a powerful analytical tool, HPLC, according to the method [31], and the results showed that dihydroquercetin (DHQ), quercetin, and kaempferol were detected in coronas and decreased with corona development (Figure 1B). All carotenoids (lutein, zeaxanthin, and ß-carotene), flavonoids (rutin, naringenin), and chlorophyll (chlorophyll a, chlorophyll b) at first increased from T1 to T2, up to the peak at T2, and then decreased in tepals during flower development. Carotenoids (lutein, zeaxanthin, and ß-carotene), flavonoids (rutin, naringenin), and chlorophyll (chlorophyll a, chlorophyll b) at first increased from C1 to C2 in coronas, then the flavonoids (rutin, naringenin) and chlorophyll were stably maintained at C2 to C3, but carotenoids (lutein, zeaxanthin, and ß-carotene) continued to increase at the C3 stage, then decreased at the C4 stage in coronas (Figure 1C). Referring to the scent profile of cut narcissus flowers [32], we detected floral fragrance components, and the results exhibited a difference between their fragrant component and emitting profile between tepals and coronas during flower development. The volatile organic components (VOCs) were not detectable at C1/T1 stage (data not shown). Benzyl benzoate, 3-hexenal and 2-hexenal were tissue-specifically detectable in tepals. Benzyl benzoate significantly increased and 3-hexenal and 2-hexenal significantly decreased from T2 to T4 in tepals (Figure 1D), while indole, benzenepropanoic acid, α-hydroxy-, methyl ester, methoxyphenyl ethylamine, phenethyl acetate, and 1,8-cineole were tissue-specifically increased in the coronas from C2 to C4, and (*E*)-3-hexenyl acetate significantly increased in the coronas from C2 to C3 and then decreased in C4 (Figure 1E), (*E*)-ocimene, benzyl acetate, heptanal, and y-n-heptyl butyrolactone were increasingly emitted both in the tepals and coronas (Figure 1F). Phenylpropyl acetate increased in tepals, maintained stable in coronas from C2 to C3, but increased in C4 (Figure 1F).

Collectively, during the flower development of *Narcissus* (Figure 1A) aromatic esters, aliphatic esters, and monoterpenes were increased in both the tepals and coronas; but aliphatic alcohols and aliphatic aldehydes decreased in the tepals from the T2 to T4 stage (Figure 2D); Simultaneously, flavonoids and carotenoids were declined in the tepals, maintained stable or increased in the coronas from the C2 to C3 stage (Figure 2B). Taken together, these observations suppose that branches of the phenylpropanoid pathway mainly leading to the production of color (white, naringenin, rutin) and fragrance (benzyl benzoate) exists in tepals (Figure 2A,C); In contrast, branches of the methylerythritol phosphate (MEP) (Geranylgeranyl diphosphate synthase, GGPS) metabolic pathways mainly leading to the production of color (orange, carotenoids and flavonoids) and scent (indole, benzyl acetate, ocimene) exist in coronas (Figure 2A,C).

### 2.2. WGCNA Analysis Displays Various Pigment and Fragrance Metabolism Enrichment Pathway during Tepal and Corona Development of Narcissus tazetta

To discover the structure of the pigment and fragrance metabolic pathways and the relationship between flower color and scent formation during flower development, a transcriptome-profiling analysis of the tepals and coronas of *Narcissus tazetta* at different developmental stages was performed. The corona transcriptomes of C1, C2, C3 and C4, and the tepal transcriptome at T0 were performed by RNA-seq in this study ((https://submit.ncbi.nlm.nih.gov/subs/SUB10083597), and the transcriptome data of the tepals at the T1, T2, T3 and T4 stages were retrieved from our previously published data [2]. We compared expression in all comparison pairs of the nine samples, and in total 15,048 genes were differentially expressed (DEGs, |logFC > 1|, FDR < 0.01). Weighted gene co-expression network analysis (WGCNA) is used for mining tissue-specific modules and key genes related to the phenotype. In order to explore key genes and co-expression networks which play important roles during the development of narcissus tepals and coronas, we analyzed 15,048 DEGs from 9 samples by WGCNA. In order to construct a scale-free network, the optimal soft threshold was set at 11. The adjacency matrix and tom overlap matrix were established by using the function adjacency and tom similarity [33]. The modules were divided based on the dynamic cutting tree, and the small modules with high similarities were merged. A total of 16 modules were obtained. In order to explore the specific modules of narcissus tepal and corona traits during flower development, a labeled heatmap function was used to visualize and analyze the relationship between the modules and samples. The module with correlation coefficient above 0.65 and *p <* 0.05 was defined as the sample-specific module, thus, 11 tissue-specific modules were obtained in 9 samples, as shown in Figure 3A. Orange4 was positively correlated with C1; lavender blush3 and plum2 were positively correlated with C2; and yellow and sky blue were positively correlated with C3 and C4, respectively. The specific modules of T0, T1, T2, T3 and T4 were steelblue, bisque4, white, brown4 and coral1, respectively.

In order to gain insight into the metabolic pathways that play important roles during the development of the tepal and corona, KEGG enrichment analysis was carried out in the genes on the above specific modules, and a significant enrichment pathway was screened (Q < 0.05). The results showed that in tepals: a “plant hormone signal transduction” pathway was significantly enriched at the T0 stage; the “photosynthesis, metabolism of purine and chlorophyll, and biosynthesis of flavonoids” pathway was enriched at the T1 stage; the “phenylpropanoid biosynthesis and aromatic compounds degradation” pathways were enriched at the T3 stage; and the “carotenoid biosynthesis” pathway was enriched at the T4 stage (Figure 3B). In coronas: C1 specific module genes were enriched in “replication and repair”, “phenylpropanoid biosynthesis”, “flavonoid biosynthesis”, “phenylalanine metabolism” and other pathways; C3 specific module genes were also enriched in the “carotenoid biosynthesis” pathway; C4 specific module genes were enriched in “phenylpropanoid biosynthesis” and other pathways (Figure 3B). Similar to the tepal, the pathways related to flower color, such as the “phenylpropanoid biosynthesis”, “flavonoid biosynthesis” and “carotenoid biosynthesis” pathways, were enriched in the coronas at the C1, C3 and C4 stages. Interestingly, numerous genes related to floral pigment biosynthesis pathways were more enriched in the coronas than in the tepals, which may be the reason for the difference of color between the tepals and coronas.

### 2.3. C4H and 4CL Transcript Levels Determines Metabolite Flux between Anthocyanin/Flavonol Biosynthesis and the Formation of Benzenoid/Phenylpropanoid Components

Both branches of the phenylpropanoid pathway, leading to the production of color and scent in flowers, have been demonstrated in petunias [8]. In order to characterize the branches of the phenylpropanoid pathway in the *Narcissus* flower, transcriptome datasets from our previous work [2] were reanalyzed. The transcript profiles of flavonoids (anthocyanin/flavonols) and biosynthesis-related enzymes in tepals and coronas during flower development were exhibited in the pathway (Figure 4), and showed that *PHENYLALANINE AMMONIA LYASE* (*PAL*) and *4CL* (c19257) were upregulated in both the tepal and corona during flower development, while *C4H* (c33958), *CHALCONE SYNTHASE* (*CHS*), and *FAVANONE 3-HYDROXYLASE* (*F3H*) were downregulated in both the tepal and corona from T1 to T4. However, *DIHYDROFLAVONOL 4-REDUCTASE* (*DFR*) and *FLS* were upregulated in corona, although FPKM values were low, one of *C4H* (c114359, orange) stably maintained a high level in coronas during flower development (Figure 4 right; Appendix A). Simultaneously, the transcript profiles of benzenoid/phenylpropanoid biosynthesis-related enzymes in tepals and coronas during flower development were exhibited in the heatmap (Figure 4 left; Appendix A), and indicated that *AMINE OXIDASE* (*AOC3*), *PHENYLACETALDEHYDE REDUCTASE* (*PAR*), and *3-KETOACYL-COA THIOLASE* (*KAT1*) genes were up-regulated both in tepals and coronas.

Summarily, in tepals with flower development, increasingly expressed *4CL* and decreasingly expressed *C4H* led to metabolite flux from the anthocyanin/flavonols biosynthesis pathway to the formation of the benzenoid/phenylpropanoid component pathway. While one of *C4H’s* expression levels remained unchanged in coronas, *DFR* and *FLS* expression levels slightly increased in coronas. Compared with the above changing profiles of pigment and fragrance component in tepal and corona during flower development, it suggests that *C4H* and *4CL* transcript levels affected metabolite flux from the anthocyanin/flavonols biosynthesis pathway to the formation of the benzenoid/phenylpropanoid component pathway, leading to an increasing amount of products of benzenoid/phenylpropanoid in the tepals and coronas, and a decreasing amount products of flavonols. Additionally, there was no anthocyanin in tepals from T1 to T4. It is similar with diversified branches of the phenylpropanoid pathway, leading to the production of color and scent in petunias [8]. Taken together, we established three branches of the phenylpropanoid pathway in the *Narcissus* flower, to flavonols/anthocyanin (color), and two pathways to scent (benzyl benzoate/phenyethyl benzoate, and benzyl acetate) (Figure 4).

### 2.4. GGPPs Transcript Levels Coordinate Metabolite Flux between Monoterpene and Carotenoids Biosynthesis Pathway in Coronas

Three branches of the MEP pathway, leading to carotenoids biosynthesis, the biosynthesis of the phytol side chain of chlorophylls, and of diterpenoids such as gibberellins, were demonstrated in many plants [34,35,36,37,38,39,40,41,42]. In tomato plants, carotenoids biosynthesis started from the MEP pathway in plastids. The MEP pathway produced the five-carbon isomers, isopentenyl diphosphate (IPP) and dimethylallyl diphosphate (DMAPP), with pyruvate and glyceraldehyde 3-phosphate as substrates. Three molecules of IPP were condensed with one molecule of DMAPP into geranylgeranyl diphosphate (GGPP) by GGPP synthase (GGPPS). GGPP is not only used for carotenoid biosynthesis. It is also the immediate substrate for the biosynthesis of the phytol side chain of chlorophylls and of diterpenoids such as gibberellins. The allocation of GGPP among different downstream metabolic branches largely determined the biosynthetic capability of each branch [42]. In *Narcissus*, orange color (a high amount of carotenoids and flavonols) and a high amount of monoterpenoids and indoles were accumulated, and a dwarfism corona phenotype was compared to the phenotype of the tepal (white in color, high in benzyl benzoate contents, and decreasing in hexenal content), supposing the competition of the carotenoid and terpene backbone biosynthetic branches for GGPP existed in the corona compared to in the tepal. The branches of the MEP pathway in the *Narcissus* flower were established by transcriptome datasets analysis, and gene expression profiles of structure genes were exhibited in the pathway (Figure 5; Appendix A). GGPP is produced by plastid GGPPS and serves as a precursor for vital metabolic branches, including chlorophylls, carotenoids, and terpene backbone biosynthesis (gibberellin biosynthesis). PHYTOENE SYNTHASE (PSY) is the entry enzyme that directs metabolite flux into carotenoids biosynthesis by condensing two molecules of GGPP into a phytoene; CHLOROPHYLL SYNTHASE (CHLG) is a main enzyme that directs metabolite flux into the biosynthesis of the phytol side chain of chlorophylls, and *ENT*-COPALYL DIPHOSPHATE SYNTHASE (CPS) is an entry enzyme that directs metabolite flux into terpene backbone biosynthesis (gibberellin biosynthesis) [42]. In *Narcissus*, with the time course of flower development, the *GERANYLGERANYL PYROPHOSPHATE SYNTHASE* (*GPPS*) and *GGPPS* transcript level was upregulated in the tepal and corona. Transcript levels of *PSY* was downregulated in both tepal and corona, *GERANYLGERANYL*
*REDUCTASE* (GGR) was downregulated in the tepal but maintained a high level in the corona, even upregulated from the C1 to C3 stage of the corona, while the transcript levels of *CPS*, *TERPENE SYNTHASE* (*TPS*), and *PAL* (Figure 4) in diterpenoids/monoterpene biosynthesis and benzenoid/phenylpropanoid pathways were significantly upregulated in both the tepal and corona during flower development. It suggests that high *CPS*/*TPS*, *PAL* and low *PSY*/*GGR* transcript levels affect metabolite flux to the terpene backbone component pathway, to the formation of benzenoid/phenylpropanoid, or to the carotenoids biosynthesis pathway during flower development, compared to the above changing profiles of pigment and fragrance component in the corona during flower development (Figure 1). It is consistence with metabolic flux among the branches of the carotenoids pathway, that leads to the production of the yellow color and increasing indole, (*E*)-β-ocimene, and benzyl acetate scent in the corona of narcissus.

### 2.5. WGCNA Analysis of Transcription Factors Related to Color and Scent Biosynthesis Enrichment Term of Narcissus Flower Development

In order to identify transcription factors (TFs) for regulating the metabolite flux to the anthocyanin/flavonols biosynthesis pathway or to the formation of the benzenoid/phenylpropanoid component pathway during *Narcissus* flower development, a total of 3193 transcription factors were identified from our transcriptome dataset tepal (Appendix A). WGCNA analysis of 3193 TFs produced 18 co-expression modules in tepals and coronas during flower development (Figure 6A), and 12 sample-specific modules (module-sample correlations > 0.65 or < −0.65, *p*-values < 0.05) were identified (Appendix A). The majority of expressed genes in sample-specific modules showed sample-preferential expression patterns as indicated in Figure 6B. Finally, these TF genes were annotated and mapped to various stages of tepal and corona tissue, and showed that NAC, WRKY, C3H, AP2, MYB-related, b-ZIP, MYB, C2H2, GARP, bHLH, GRAS, HB, FAR1, SBP, Trihelix, and the B3 family were involved in tepal and corona development; 24 top transcription factor families were involved in the C4 and T0 stage; LIM family was specifically involved in the C4 stage; and specifically the LOB family was involved in T0 and T4 stages (Figure 6C).

The result of Gene Ontology (GO) enrichment analysis of the transcription factor genes of sample-specific modules was exhibited in Figure 6D, and showed that anthocyanin/flavonoids biosynthesis, aroma biosynthesis, the biosynthetic process of organic nitrogen compounds, and organic cyclic compound biosynthetic process terms, were enriched in C3 (black and dark turquoise module) and T3 (pink module) (Figure 6D and Appendix A).

In order to avoid losing multiple functional TFs, we performed a second-round alterative screen. To this end, WGCNA analysis of DEGs was performed to find the co-expression modules of pigment and fragrance metabolism and to identify the TFs among them. According to the role that the highly connected hub gene is usually a regulatory factor (an upstream position in a regulatory network), generally, the KME (eigengene connectivity) value is calculated and the hub gene is selected by using | KME | > = threshold (0.8), Top100 or 30%; the selected hub genes possess high connectivity in modules [43]. The result of KEGG enrichment analysis of sample-specific modules exhibited that carotenoid, flavonoid, chlorophyll biosynthesis-related genes and terpenoids, and benzenoid/phenylpropanoid biosynthesis-related genes were enriched in C1 orangered4, C3 yellow, T1 bisque4, T3 brown4, T4 coral1 five floral pigment and fragrance formation-specific gene co-expression modules (Q < 0.05) (Figure 3B). Among the top 30% hub genes related to floral pigment and floral fragrance formation in petals and coronas during flower development, 130 TFs, including AP2/ERF, B3, BBR, BES1, bHLH, bZIP, C2H2, C3H, E2F, GARP, HB, HSF, LOB, MADS, MYB, NAC, NF, OFP, PLATZ, RWP, S1Fa, SBP, SRS, TCP, Trihelix, WD40 and the ZF families, were identified in the C1 module of first developmental stage of coronas. Fourteen TFs, including AP2-ERF, bZIP, GARP, HB, LOB, MYB, NAC, PLATZ, WD40, and the ZF families, were identified in the C3 module of the developmental stage of coronas. Thirty-six TFs, included in AP2/ERF, B3, BBR, bHLH, bZIP, C2C2, C2H2, C3H, MYB, NAC, NF, OFP, SiFa, SBP, TCP and the ZF families, were identified in the T1 bisque4 module of first developmental stage of tepals. Sixty TFs, including AP2/ERF, B3, bHLH, bZIP, C2C2, C2H2, E2F, GARP, GRAS, HB, MYB, NAC, NF, OFP, PLATZ, Tify, ULT, WD40, WRKY, and the ZF families, were identified in the T3 brown4 module of the later developmental stage of the tepals. Sixty TFs, including AP2-ERF, bHLH, bZIP, C2C2, C2H2, DBP, E2F, GARP, GRAS, HB, HSF, LOB, MYB, NAC, NF, Trihelix, ULT, and the WRKY families, were identified in the T4 coral1 module of the later developmental stage of tepals (Appendix A).

### 2.6. Identification of Transcription Factors Related to Color and Scent Biosynthesis during Narcissus Flower Development

In order to identify more specific TFs related to floral pigment and floral fragrance formation during *narcissus* flower development, we made KEGG enrichment analysis of TFs of sample-specific modules exhibited in Appendix A, and interestingly showed that the T3 brown4 module of the later developmental stage of petals appeared to significantly show aroma biosynthesis enrichment and organic substrate biosynthesis enrichment terms. Thereby, four candidate TFs genes of 60 transcription factors at the T3 stage, including NF transcription factors, and 20 transcription factors from the top 30% hub genes of modules related to flower color and floral fragrance, were selected according to GO terms related to the “aroma biosynthesis” enrichment term, “organic substrate biosynthesis” enrichment, “cellular biosynthetic process”, “carbohydrate metabolic process”, and “regulation of transcription”. The C3 dark turquoise4 and black modules of the later developmental stage of coronas appeared significantly in the “anthocyanin biosynthesis in expose to UV light” and “organic substrate metabolism” enrichment terms; four other transcription factor genes of 14 transcription factors at C3 stage were selected. A total of 28 significantly enriched TFs of 354 detected TFs were exhibited in heatmap, including NF, MYB, B3, SBP, NAC, C3H, C2C2, C2H2, GRAS, HB, AP2/ERF, BEST1, TUB, and WRKY family (Figure 7A).

Furthermore, theses 28 TFs (Figure 7A) were taken to perform a co-expression analysis with structural genes related to aroma biosynthesis and organic substrate biosynthesis (the enrichment term in the top 30% hub genes of modules was related to floral pigment and floral fragrance during flower development by Cytoscape) [44]. The results showed that *MYB*, *NAC*, and *AP2-ERF* coexpressed closely with the top five genes, *4CL, C4H*, and *FLS,* that were involved in metabolite flux from anthocyanin/flavonols biosynthesis to the formation of the benzenoid/phenylpropanoid component (Figure 7B). *AP2-ERF, bZIP, GARP, NF, bHLH, SBP*, and *B3* shared a coexpression with the *GLUTAMATE-1-SEMIALDEHYDE 2,1-AMINOMUTASE* (*GSA*) gene which encoded HemL in the chlorophyll metabolism pathway (Figure 7C); *bZIP* and *GRAS* coexpressed with *DXR*, *PDS* and *NCED* genes in the carotenoid metabolism pathway (Figure 7D); *C2C2* and *C2H2* coexpressed with *POR* genes in chlorophyll a biosynthesis pathway (Figure 7E).

In order to genetically evaluate potential regulation, we compared the expression trend of candidate regulatory TFs between the tepals of *Narcissus pseudonarcissus* cultivated variety “European yellow narcissus” relative to white tepals of *Narcissus tazetta* cultivated variety “Jinzhanyintai”, and the yellow coronas of *Narcissus tazetta* relative to white tepals of *Narcissus tazetta*. We searched for RNA-seq dataset for the yellow tepals of *Narcissus pseudonarcissus* species (unpublished data). Among 20 candidate TFs from regulatory networks (Figure 7B–E), seven TFs including *bZIP*, *GRAS*, *C2C2*, *C2H2*, *NAC*, *MYB*, *AP2-ERF* had the same up/down regulated trend in coronas and tepals in *Narcissus pseudonarcissus* compared with the white tepals of *Narcissus tazetta* (Figure 8), especially, in the carotenoid biosynthesis pathway (Network E), *GRAS* and *bZIP* transcription factor genes had same expression pattern with *NCED*, *DXR* and *PDS*. In the flavonoids biosynthesis pathway, or in the benzenoid/phenylpropanoid biosynthesis pathway, *NAC* and *AP2-ERF* transcription factors had the same expression pattern as *FLS* and *C4H.* While *MYB12* was upregulated in *Narcissus pseudonarcissus*, *MYB1*, *MYB* (c65310) and *4CL* were upregulated in the coronas of *Narcissus tazetta*, hinting that there was a different metabolic flux direction between the color and scent component formation in *Narcissus pseudonarcissus* and *Narcissus tazetta*. It suggested that the potential competition of a regulatory network exists between floral pigment and floral fragrance during the individual plant development of *Narcissus* and evolutionary development.

## 3. Discussion

From a genetic and metabolic viewpoint, the most extensively studied flower color and flower scent is that of the venation. Studies in petunia (*Petunia hybrida*) have revealed a conserved mechanism for the formation of the benzenoid/phenylpropanoid component and flavonol/anthocyanin pigmentation in the petals. In this study, we used an individual plant and two similar tissues (tepal and corona). To create an isogenic line, we established a regulatory network of three branches of the phenylpropanoid pathway and three branches of the MEP pathway of *Narcissus tazetta* by using transcriptome-based WGCNA analysis. Our results highlight the potential metabolic flux of substrate competition in generating spatial patterns with color and scent contrast, and provide several candidate transcription factors: MYB12, MYB1, AP2-ERF, bZIP, NAC, MYB, C2C2, C2H2, and GRAS, which are associated with metabolite flux. It indicates that competition between benzenoid/phenylpropanoid and flavonol/anthocyanin biosynthesis produces white and aromatic tepals of narcissus, and metabolic flux among monoterpenoids and chlorophyll. The carotene biosynthesis pathway produces orange and protective aromatic coronas of *Narcissus tazetta*.

In *Narcissus*, main floral pigment (carotenoids and flavonoids), and floral fragrance (benzenoid/phenylpropanoid) was provided [31,32]. A cultivated variety “Jinzhanyintai” of *Narcissus tazetta* var. chineses, with its variation of floral scent and floral color (the white tepal and yellow corona have a different pigment and fragrance component) might simply be explained by floral color change, but may also be influenced by multiple and/or conflicting selection pressures acting on the pleiotropic effects of flower traits. Whether pollinator-imposed selection may have a direct impact on floral scent emissions, independent of biochemical constraints due to floral color, remains unanswered. Increasing research on the effects of floral color and scent on pollinator attraction have been investigated in different species [45]. Odor signals from flowers have been hypothesized to play a minor role in pollinator orientation towards various color, e.g., purple or white morphs [45]. The evolution of angiosperms has resulted in an immense diversity of flower traits such as shape, size, color, and scent. Interestingly, the quality and quantity of emitted volatiles are species-specific and tissue-specific among different populations of a given species [46]. Evolution of floral scent is potentially shaped by two factors that mutually influence each other: (1) genomic changes allowing catalytic expansion and differential regulation of the enzymatic machinery underlying floral scent and floral color formation (Figure 4, Figure 5 and Figure 8) and (2) ecological constraints such as pollinator-mediated selection. Thus, white color *Narcissus* tepal organs possess a rich scent or a colorful *Narcissus* corona organ is scentless (Figure 1 and Figure 2).

TFs controlling the metabolic flux among the phenylpropanoid/benzenoid network was reported in petunia flowers. ODORANT1 (ODO1), a R2R3-type MYB TF, was exclusively expressed in petunia petal tissue and regulated the transcription of a major portion of the shikimate pathway, as well as entry points into both the Phe (i.e., chorismate mutase) and phenylpropanoid (i.e., PAL) branch ways [21]. The gene *ODO1* was positively regulated by another R2R3-type MYB TF, EMISSION OF BENZENOIDS II (EOBII) and EOBI, which also activated the promoter of the biosynthetic gene, isoeugenol synthase [22,47,48]. In contrast to ODO1, EOBI, and EOBII, the PhMYB4 TF was found to be a repressor of only a single enzyme in the benzenoid/phenylpropanoid pathway, cinnamic acid 4-hydroxylase, thus controlling the flux towards benzenoid/phenylpropanoid volatile compounds in petunia flowers [23,30]. In the narcissus plant, it was reported that NtMYB3, an R2R3-MYB from *Narcissus*, which belongs to the AtMYB4-like clade, was involved in the regulation of flavonoid biosynthesis in narcissus by repressing the biosynthesis of flavonols, and this led to proanthocyanin accumulation in the basal plate of narcissus. However, it seemed to be more similar to a flower-specific MYB homologous to *Arabidopsis* MYB21 that regulated stamen development and pollen phenylpropanoid and was involved in elevating the *NtFLS* transcript levels [49,50]. Therefore, NtMYB3 might be mapped to a later step of the flavonoids biosynthesis pathway. Thus, it did not appear in our regulatory network. However, NtMYB12, a homologous protein of AtMYB12 and PhMYB4, played a role in the activation function of flavonols in *Arabidopsis* and *Petunia* [30,51]. The NtMYB1, a homologous protein of AtMYB75 and AtMYB4, had an MYB transcription factor regulating both carotenoid and gibberellin biosynthesis pathways that was reported in *Arabidopsis* and tomato plants [42]. Although it is clear that MYB is one of the main transcription factors regulating the formation of flower color, most members of MYB family positively regulate the synthesis of flavonoids [52], in which MYB needed bHLH and WD40 to participate in the formation of the MBW complex in the regulation of anthocyanin synthesis [53,54,55]. The NtNAC78-like, is homologous to the *Arabidopsis* NAC Transcription Factor, ANAC078, which regulated flavonoid biosynthesis via PAP1 under high-light [56,57]. The NtAP2-ERFwas a homologous of AtAP2-ERF and CsAP2-ERF, which was involved in controlling the expression of *BENZOIC ACID/SALICYLIC ACID CARBOXYL METHYLTRANSFERASE* (*PhBSMT1* and *2*), where mRNA is temporally and spatially down-regulated in floral organs in a manner consistent with reported models for post-pollination ethylene synthesis in petunia corollas [58,59]. In addition, there are many other transcription factors such as WRKY, bZIP and MADS, involved in the regulation of anthocyanin in *Arabidopsis*. The bZIP is involved in the regulation of the phenylpropane metabolism pathway [58,59]. In this study, although five candidate TFs such as bZIP(c124715), SBP (c97416), B3(c120916), SBP (c107934), and bHLH(c119417) could not be identified from the sequencing dataset. This was perhaps due to the different genotypes in *Narcissus tazetta* and *Narcissus pseudonarcissus*. The opposite expression trends of *NF* (c105468) and *HSF* (c91958) in the chlorophyll metabolism pathway of yellow tepals, compared with the coronas of *Narcissus tazetta*, may have resulted from their multiple functions during development. Most of the candidate TFs, including MYB12, bZIP, MYB1, MYB, C2C2, C2H2, GRAS, NAC, and AP2-ERF had the same changing trends confirmed in the tepal of *Narcissus pseudonarcissus* (Figure 8). Therefore, these regulators, including MYB1, MYB, MYB12, bZIP, NAC, C2C2, C2H2, GRAS, and AP2/ERF, which orchestrated the formation of diverse volatile blends and pigment components and acted upstream of multiple metabolic pathways, are speculative [60] (Figure 7). It remains unknown whether promoters of genes involved in the formation of phenylpropanoids and carotenoids are the natural targets for these TFs. A detailed insight into their mechanism will be the objective of future research.

## 4. Materials and Methods

### 4.1. Plant Materials

*Narcissus tazetta* cultivated variety “Jinzhanyintai” was purchased from a local farmer (Zhangzhou, Fujian, China) in October 2015 and 2020. *Narcissus pseudonarcissus* cultivated variety “European yellow narcissus” was purchased from Holland in October 2015. After removing old bulb scale and roots, cleaned bulbs were sand-cultured in a greenhouse set at 23 °C with constant illumination (16 h light/8 h dark) (Ren et al., 2017; Figure 1A). Tepals were first collected at the fifth day (sample T1) as sheathed leaves emerged from the bulbs [2] (Figure 1A), then collected at another three stages (samples T2–T4) as umbel in spathe developed [2]. All samples were immediately frozen in liquid nitrogen and stored at −80 °C.

### 4.2. Measurement of Contents of Flavonoids and Carotenoids Metabolites in Tepals and Coronas by HPLC Analysis

Tepals and cornas from S1 to S4 stages of *Narcissus tazetta* were collected for analysis the contents of flavonoids and carotenoids. HPLC were performed to analysis the extracts of flavonoids metabolism. Briefly, freeze-dried tepals and coronas (~100 mg) were homogenized and to the homogenate 1 mL of MeOH/DMSO mix (1:1, *v*/*v*) was added for 30 min at 40 °C with agitation (190 rpm) in darkness. Then samples were centrifuged at 40 °C, 9000 rpm for 15 min, the supernatants were collected and the pellets were reextracted for another 4 times until the color changed into white. All the supernatants were combined and filtered through a 0.22 μm membrane filter. The HPLC was carried out according to a previous method on an Agilent 1260 HPLC system [2]. A final volume of 10 μL was used for injection into HPLC.

For extraction of carotenoids, the method described by Ren et al. [2] was referenced with some modifications. Briefly, freeze-dried plant sample (~150 mg to 200 mg) was were finely grounded in liquid nitrogen and to the homogenate 1.5 mL of chloroform: dichloromethane (2:1, *v*/*v*) was added. The resultant suspension was sonicated for 10 min and then mixed for 10 min using a thermomixer at 1000 rpm at 4 °C. Thereafter for phase separation, 0.4 mL of 1 M sodium chloride solution was added and contents were mixed by inversion. After centrifugation at 5000 rpm for 10 min the lower organic phase was collected. The aqueous phase was re-extracted with 1 mL of chloroform: dichloromethane (2:1, *v*/*v*) and 0.2 mL of 1 M sodium chloride solution. Both organic phases were combined, dried by centrifugal evaporation, re-dissolved in 0.5 mL of methyl tert-butyl ether (MTBE, containing 0.01% BTH) and filtered through a 0.22 μm membrane filter. The HPLC was performed according to YMC separation technology-Carotene and xanthophylls protocol with minor modifications on the Agilent 1260 HPLC system (Agilent Technologies, CA, USA). A final volume of 8 μL was used for injection into HPLC.

The flavonoids and carotenoids standards including naringenin, dihydroquercetin, kaempferol, quercetin, lutein, and β-carotene were purchased from Sigma-Aldrich (St. Louis, MO, USA), rutin and zeaxanthin were purchased from TCI (TCI Co., Ltd, Japan) and Sangon (Shanghai, China), respectively. All flavonoids and carotenoids standards were prepared in MeOH/DMSO (1:1, *v*/*v*) and MTBE (containing 0.01% BHT), respectively, and stored at −20 °C before use.

### 4.3. Measurement of Chlorophyll and the Total Carotenoids Contents in Tepals and Coronas

50 to 100 mg fresh weight of tepals and coronas from S1 to S4 stages were harvested. Pigments were extracted by immersing sample in 1 mL of 95% ethanol (*v*/*v*) and kept at 4 °C in dark for 48 h until the tissues became white. After centrifugation at 15,000 rpm for 5 min, the supernatant was collected and was used to measure the absorbance at 470, 649, and 665 nm in 96-holes ELISA plate by using a Flexstation 3 Microplate Reader (Berthold, BadWildbad, Germany). Each hole was filled with 200 μL extract and pure 95% ethanol was used as a control. Chlorophyll a, b, the total chlorophyll and the total carotenoids contents were calculated according to the method described by Ren et al. (2017) [2]. 

### 4.4. Measurement of VOCs Component in Tepals and Coronas

Sampling was performed during the winter of 2020 and 2021. The extraction and detection of VOCs was performed according to [32] with minor modification. Tepals and coronas were separated from flowers without any visual damage (from S2 to S4 stage) and grounded into fine powder in liquid nitrogen immediately. Approximately 0.4 g of the powder was extracted with 1.5 mL hexane containing 1.1 ng/μl of isobutylbenzene as the internal standard. Following a 4 h incubation with shaking at 100 rpm, extracts were centrifuged at 12,000 rpm for 10 min, and the supernatant was further centrifuged and combined, then through a 0.22 μm filter prior to chromatography. GC-MS (1 μL sample) analysis was carried out using a Clarus 680 GC with SQ8TGC/MS system (PerkinElmer, Waltham, MA, USA) coupled to a HP-5 MS capillary column (Restek; i.d. 0.25 μm, 30 m × 0.25 mm). The injection temperature was set to 250 °C (splitless mode) and the interface to 240 °C, and the ion source was adjusted to 200 °C. Helium was used as the carrier gas at a flow rate of 0.9 mL/min. The analysis was performed under the following temperature program: 2 min of isothermal heating at 40 °C followed by a 10 °C/min oven temperature ramp to 250 °C. The system was equilibrated for 1 min at 70 °C before injection of the next sample. Mass spectra were recorded at 3.15 scan/s with a scanning range of 40–500 mass-to-charge ratio and electron energy of 70 eV.

Identification and Quantification: The retention indices (RIs) were calculated by the retention time (RT) of n-alkane standards (C7–C30) under the same conditions. Floral VOCs were identified by the RIs and mass spectra compared with the reference standards in the NIST 2011 mass spectra library implemented in the Clarus SQ 8T platform. The VOCs content was calculated by normalizing the peak areas relative to the internal standard.

### 4.5. PacBio Iso-Seq Library Preparation and Sequencing

The samples of flower equally mixed from S0-S4 stage of *Narcissus tazetta* were collected for PacBio full-length sequencing, of which all of the raw data, including Pacbio full-length-seq and former 4 DGE-seq library pools (GenBank Short Read Archive, Accession SRP083093), were used to assemble the transcriptome. All of the sequencing services were provided by Bi-omarker Technologies Co., Ltd. (Beijing, China).

The sequencing library was prepared according to the Iso-Seq protocol as described by Pacific Biosciences (P/N100-377-100-05 and P/N100-377-100-04). The SMARTer PCR cDNA Synthesis Kit was used to synthesize cDNA. After 23 cycles of PCR amplification, products were size-selected using the BluePippin Size Selection System with the following bins for each sample: 1–2 kb, 2-3 kb, and 3–6 kb. The amplified cDNA products were used to generate SMRTbell Template libraries according to the Iso-Seq protocol. Libraries were prepared for sequencing by annealing a sequencing primer and adding polymerase to the primer-annealed template. The polymerase-bound template was bound to MagBeads and sequencing was performed on a PacBio RSII instrument. SMRT-Analysis software (https://github.com/ben-lerch/IsoSeq-3.0/) was used for Iso-Seq data analysis.

### 4.6. RNA-Seq and Data Analysis

Total RNA was extracted from frozen coronas and tepals at different stages and mRNA libraries were constructed for each sample. The corona transcriptomes at stages C1, C2, C3, and C4, and the tepal transcriptome at T0, were sequenced using the Illumina HiSeqTM-2500 platform with PE150 bp model. The tepal transcriptome data of Chinese narcissus at stage T1, T2, T3, and T4, were retrieved from our published data [2] (GenBank Short Read Archive, Accession SRP083093) by RNA-seq in this study. All raw data were deposited in the GenBank Short Read Archive ((https://submit.ncbi.nlm.nih.gov/subs/SUB10083597). Clean data were obtained by removing reads containing adapters, reads containing poly-N, and low-quality reads. These clean reads were then mapped to the reference transcriptome sequence using software [61]. Only reads with a perfect match or one mismatch were further analyzed and annotated based on the reference transcriptome. The mapped fragments for each gene were counted by feature Counts [62] and fragments per kilobase per million mapped reads (FPKM) were calculated. Differential expression analysis between two samples was performed using the EBSeq R package [63]. The resulting P values were adjusted using the Benja-mini and Hochberg’s approach for controlling the false discovery rate [63]. Genes identified by EBSeq with FDR ≤ 0.01, FC ≥ 2, or ≤ 0.5, were defined as differentially expressed. All raw data were deposited in the GenBank Short Read Archive (https://submit.ncbi.nlm.nih.gov/subs/SUB10083597).

### 4.7. Identification of Co-Expression Modules and Visualization of Gene Expression

A gene coexpression network was built using the WGCNA package (Version 4.0.2). Parameters were set up as power = 12, minModuleSize = 30, and MEDissThres = 0.25. The networks were visualized using Cytoscape v.3.6.048. For high-throughput display of the expression levels of the assigned genes, heat maps were created. Related gene expression values were retrieved from databases [2]; the heat maps were plotted with the p heatmap package in R. The number of “terpenoid skeleton biosynthesis”, “monoterpenoid biosynthesis” and “sesquiterpene biosynthesis pathways” relating to plant aromas in KEGG database are ko00900, ko00902, and ko00909, respectively. The number of the “carotenoid biosynthesis”, “phenylpropanoid biosynthesis”, and “flavonoids biosynthesis” related to the formation of plant color in KEGG database were ko00906, ko00940 and ko00941, respectively. The information of the enzyme names in each of the above metabolic pathways was obtained in KEGG database, and then the KEGG map of *Narcissus tazetta* was sequenced by using the corresponding transcription group of ko number. The ID of the *Narcissus tazetta* gene annotated to the enzyme gene was found in the metabolic pathway, and the names of the genes of the related enzymes and the ID of the *Narcissus tazetta* gene were sorted. Combined with the metabolic pattern map and the expression of the enzyme gene, the heat map of the genes related to the metabolism pathway of the narcissus flower and color was drawn by TBtools after all the genes with less than 1 expression values were removed.

### 4.8. Gene Expression Validation

Expression of representative unigenes in each metabolic pathway was confirmed by semi-qRT-PCR (semi-quantitative reverse transcription and polymerase chain reaction) with specific primers designed by Beacon DesignerTM (Ver. 7, Palo Alto, CA, USA) [2]. The process of cDNA synthesis was done by using RevertAid First Strand cDNA Synthesis Kit (Thermo Scientific, Waltham, MA, USA) following the manufacturer’s protocol. Semi-qRT-PCR was performed on an ARKTIK thermal cycler (Thermo Scientific). The unigene annotated to GAPDH (c28660.graph_c0) was chosen as an internal control (25 cycles). Three independent biological repetitions with three technical replicates were performed. PCR products were run on a 2.0% TAE agarose gel and calculated by measuring the grey level of each lane using Alpha View software (San Jose, CA, USA) (Yang Jingwen Master thesis, 2016, Fujian Agriculture and Forestry University). 4.9. Statistical Analyses

### 4.9. Statistical Analyses

A one-way analysis of variance (ANOVA) in combination with Duncan’s multiple range test or Kruskal–Wallis test with a significance of differences of *p* < 0.05 were conducted by data processing system (DPS) version 7.05 software (Stirling Technologies Inc., China). Means indicate at least three biological replicates. The heatmap is drawn by TBtools software (CJ Chen, Guangzhou, China). 

## Figures and Tables

**Figure 1 ijms-22-08249-f001:**
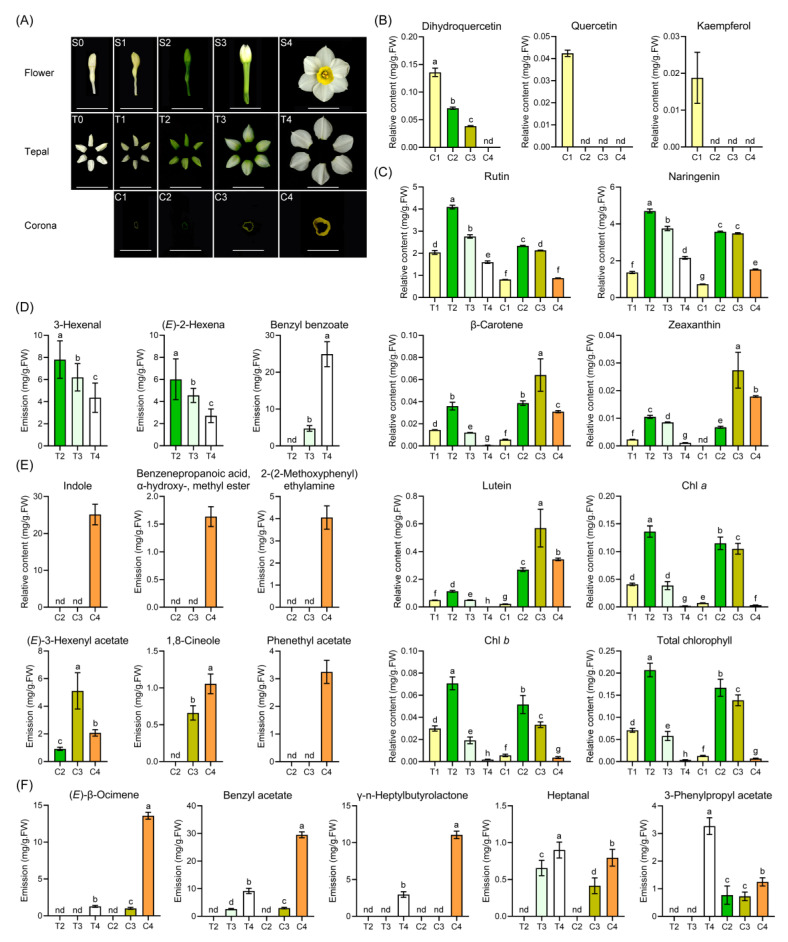
Changes of floral pigment and fragrance components in tepals and coronas during *narcissus* flower development. (**A**) Development process of *Narcissus tazetta* flower. (**B**) Changes of floral pigment during corona development (C1–C4) (**B**). (**C**) Changes of floral pigment during tepal (T1-T4) and corona development (C1–C4). (**D**–**E**) Changes of fragrance in tepal (**D**) and corona (**E**) during narcissus flower development. (**F**) Changes of floral fragrance in both the tepals and coronas during narcissus flower development. The fragrance at C1 and T1 stage were not detectable. All data are from at least three or more biological replicates and expressed as means ± SD. Letters “a”, “b”, ”c”, ”d”, ”e”, ”f” and “g” indicate statistically significant differences for the indicated values, as determined by a one-way analysis of variance (ANOVA) with Duncan’s multiple range test and Kruskal-Wallis test (*p* < 0.05), nd, not detected.

**Figure 2 ijms-22-08249-f002:**
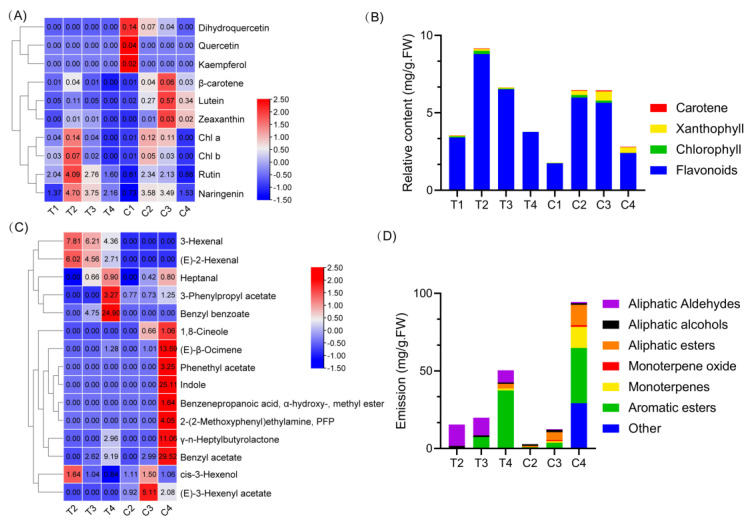
Comparison of the changing profiles of floral pigment and fragrant component in tepals and coronas. (**A**). The heatmap shows the amount of pigments in tepals and coronas during flower development. (**B**). Changing profiles of pigments in tepals and coronas; (**C**). The heatmap shows volatile organic components (VOCs) amounts in tepals and coronas during flower development; (**D**). Changing the profiles of volatiles in tepals and coronas. The heatmap was drawn by TBtools software and the function “scale” was used for normalized the data by rows.

**Figure 3 ijms-22-08249-f003:**
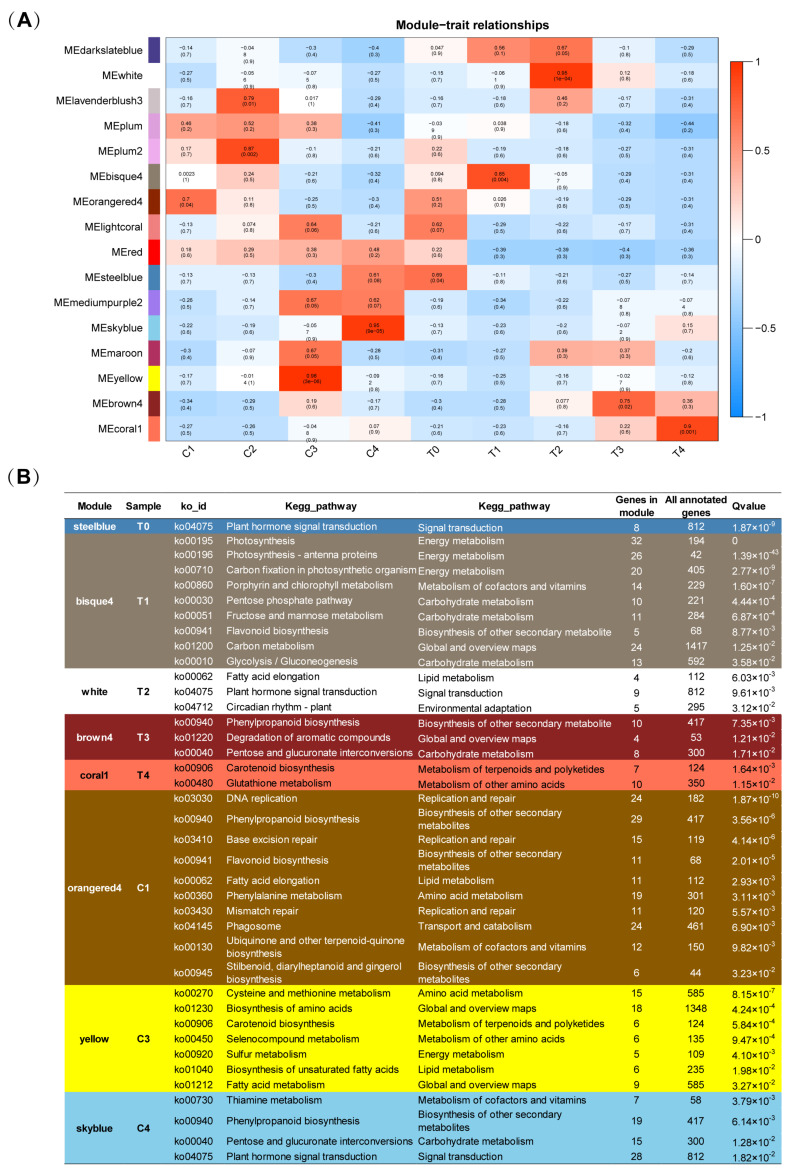
Weighted Gene Co-Expression Network Analysis (WGCNA) for differentially expressed genes in *Narcissus* flowers and KEGG enrichment analysis of sample-specific modules for different development periods of coronas and tepals. (**A**). WGCNA analysis for differentially expressed genes of *Narcissus tazetta*. Corresponding *p*-values of module-sample correlations are indicated in parenthesis. The panel on the left side shows the sixteen modules. The color scale on the right side shows module-trait correlation from −1 (blue) to 1 (red). (**B**). KEGG enrichment analysis of sample-specific modules for different development stages of tepals and coronas.

**Figure 4 ijms-22-08249-f004:**
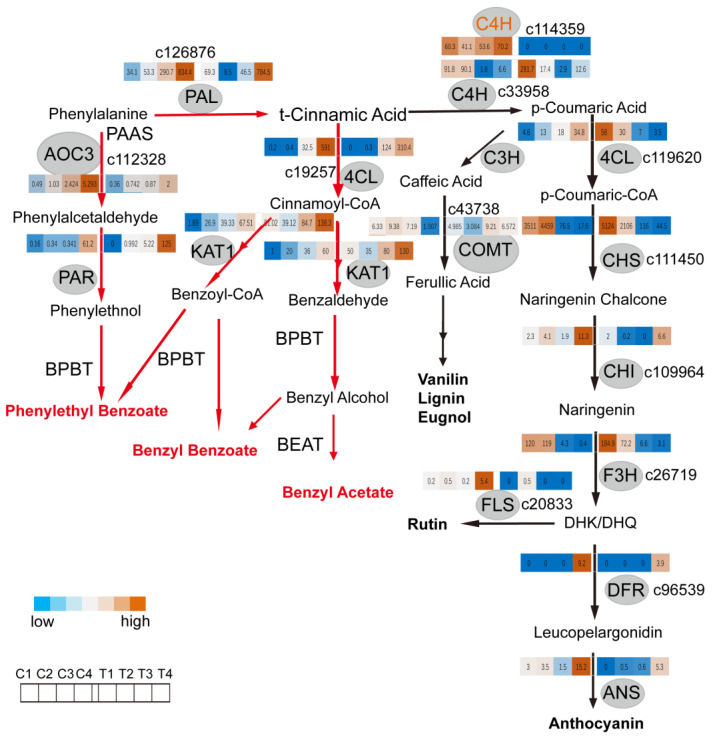
Expression patterns of DEGs related to branches of the phenylpropanoid pathway in tepals and coronas of *Narcissus* during flower development. Branches of the phenylpropanoid pathway in *Narcissus* and expression patterns analysis of DEGs, are related to flavonoids metabolic pathway and benzenoid/phenylpropanoid metabolic pathway. The red line and arrowhead in the left panel represent the phenylpropanoid biosynthesis pathway, and the black line and arrowhead represent the flavonoid/anthocyanin biosynthesis pathway. The heat map shows the expression patterns of differentially expressed structural genes related to branches of the phenylpropanoid pathway in the tepals and coronas at different periods (left down). Each row means one unigene and the expression changes of unigenes are shown by color grids based on FPKM value (fragments per kilobase of exon per million reads mapped). The color bar from blue to orange shows upregulated; from orange to blue is downregulated. Enzyme names are shown in the gray background circle. 4CL, 4-coumarate CoA ligase; KAT, 3-ketoacyl-CoA thiolase; C4H, cinnamic acid 4-hydroxylase; AOC3, amine oxidase, copper containing 3; PAR, phenylacetaldehyde reductase; BEAT, acetyl-CoA: benzylalcohol acetyltransferase; BPBT, benzoyl-CoA: benzoyl-CoA: benzyl alcohol/phenylethanol benzoyltransferas; PAL, phenylalanine ammonia-lyase; PAAS, phenylacetaldehyde synthase; CHS, chalcone synthase; CHI, chalcone isomerase; F3H, favanone 3-hydroxylase; FLS, flavonol synthase; DFR, dihydroflavonol 4-reductase; ANS, anthocyanin synthase; C3H, p-coumarate 3-hydroxylase; COMT, caffeic acid 3-o-methyltransferase.

**Figure 5 ijms-22-08249-f005:**
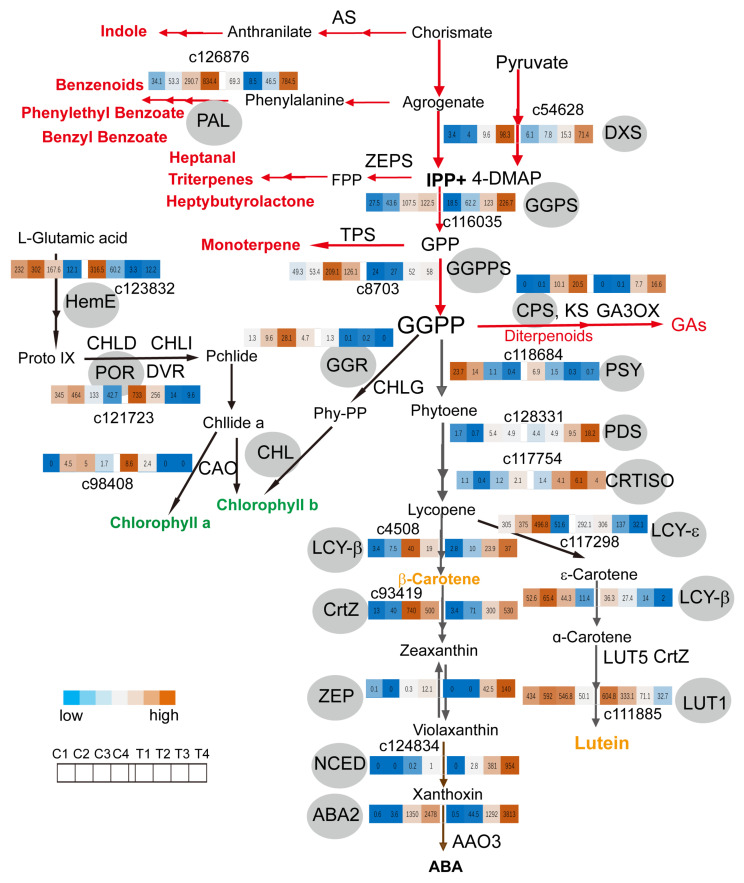
Three branches of MEP pathway include carotenoids biosynthesis, the biosynthesis of the phytol side chain of chlorophylls, and of diterpenoids such as gibberellins in the *Narcissus* flower. The red line and arrowhead in the up panel represents the terpenoids metabolic pathway, and the black line and arrowhead represents carotenoids biosynthesis; the biosynthesis of the phytol side chain of the chlorophylls pathway. The heat map shows the expression patterns of differentially expressed genes related to three branches of MEP pathway in tepals and coronas at different periods (left down). Enzyme names are shown in the gray background circle. The FPKM values of genes are labeled on the heat map. The color bar from blue to orange is upregulated; from orange to blue is down regulated: AS, anthranilate synthase; ZFPS, cis,cis-farnesyl diphosphate synthase; DXS, 1-deoxy-D-xylulose-5-phosphate synthase; DXR, 1-deoxy-D-xylulose 5-phosphate reductoisomerase; GGPP, geranylgeranyl diphosphate; GGPPS, geranyl diphosphate synthase; TPS, terpene synthase; PSY, phytoene synthase; PDS, phytoene desaturase; ZDS, zeta-carotene desaturase; CRTISO, carotene isomerase; LCY-ε, lycopene epsilon-cyclase; LCY-β, lycopene beta-cyclase; CrtZ, beta-carotene 3-hydroxylase; LUT5, beta-ring hydroxylase; LUT1, carotenoid epsilon hydroxylase; ZEP, zeaxanthin epoxidase; VDE, violaxanthin de-epoxidase; NCED, 9-*cis*-epoxycarotenoid dioxygenase; ABA2, xanthoxin dehydrogenase; AAO3, abscisic-aldehyde oxidase; GGR, geranylgeranyl reductase; HemL, glutamate-1-semialdehyde 2,1-aminomutase; HemE, uroporphyrinogen decarboxylase; ChlD, magnesium chelatase subunit D; CHLI, magnesium chelatase subunit I; POR, protochlorophyllide reductase; CAO, chlorophyllide a oxygenase; CLH, chlorophyllase; DVR, divinyl reductase; CHLG, chlorophyll/bacteria chlorophyll a synthase; CPS, *ent*-copalyl diphosphate synthetase; KS, *ent*-kaurene synthase; GA20OX, gibberellin 20-oxidase; GA3OX, gibberellin 3-oxidase.

**Figure 6 ijms-22-08249-f006:**
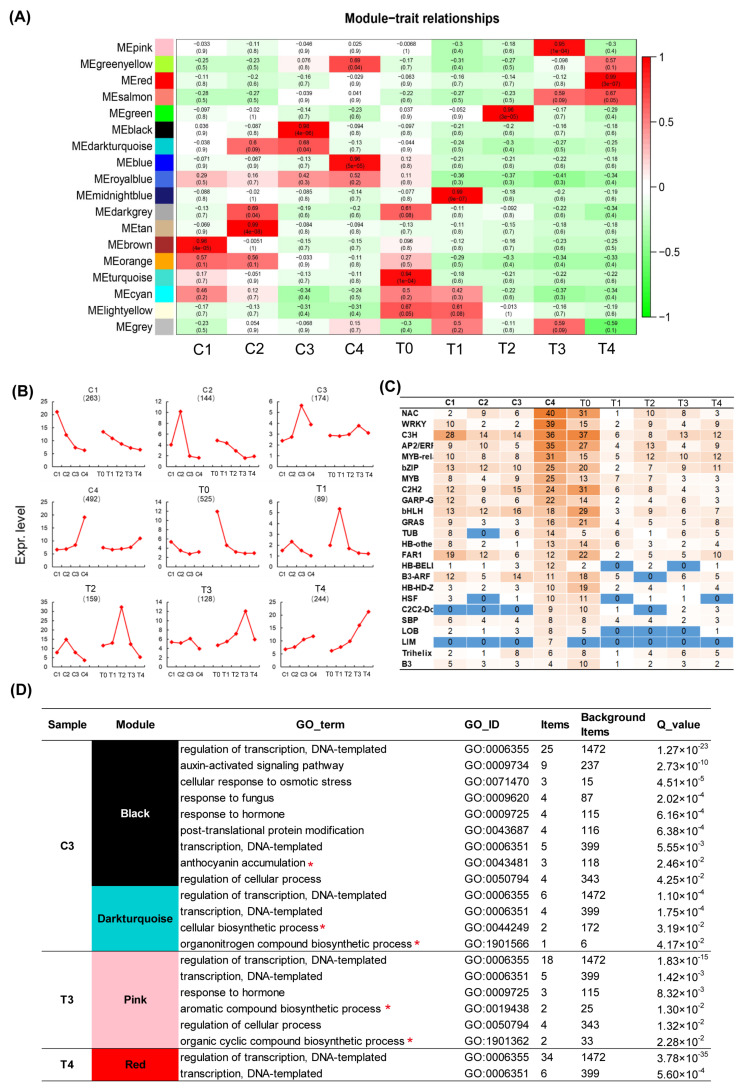
Analysis of transcription factor genes in the tepal and corona of narcissus. (**A**). Analysis of WGCNA of 3,193 transcription factor family genes. Module-sample correlations and corresponding *p*-values (in parenthesis). The left panel shows the sixteen modules. The color scale on the right shows module-trait correlation from −1 (green) to 1 (red). (**B**). Gene expression patterns within different sample-specific modules. Y axis represents the average expression value of genes in sample-specific modules, and the numbers in parentheses represent the number of genes in sample-specific modules. (**C**)**.** Distribution of transcription factor families in the tepal and corona during flower development. (**D**)**.** Functional GO enrichment analysis of the transcription factor specific module of C3, T3 and T4 refered to Figure 6A; red star labeled indicates floral pigment and fragrance metabolism related pathway. Black and darkturquoise modules are the specific modules of C3, pink is T3 sample-specific module and red is T4 sample-specific module.

**Figure 7 ijms-22-08249-f007:**
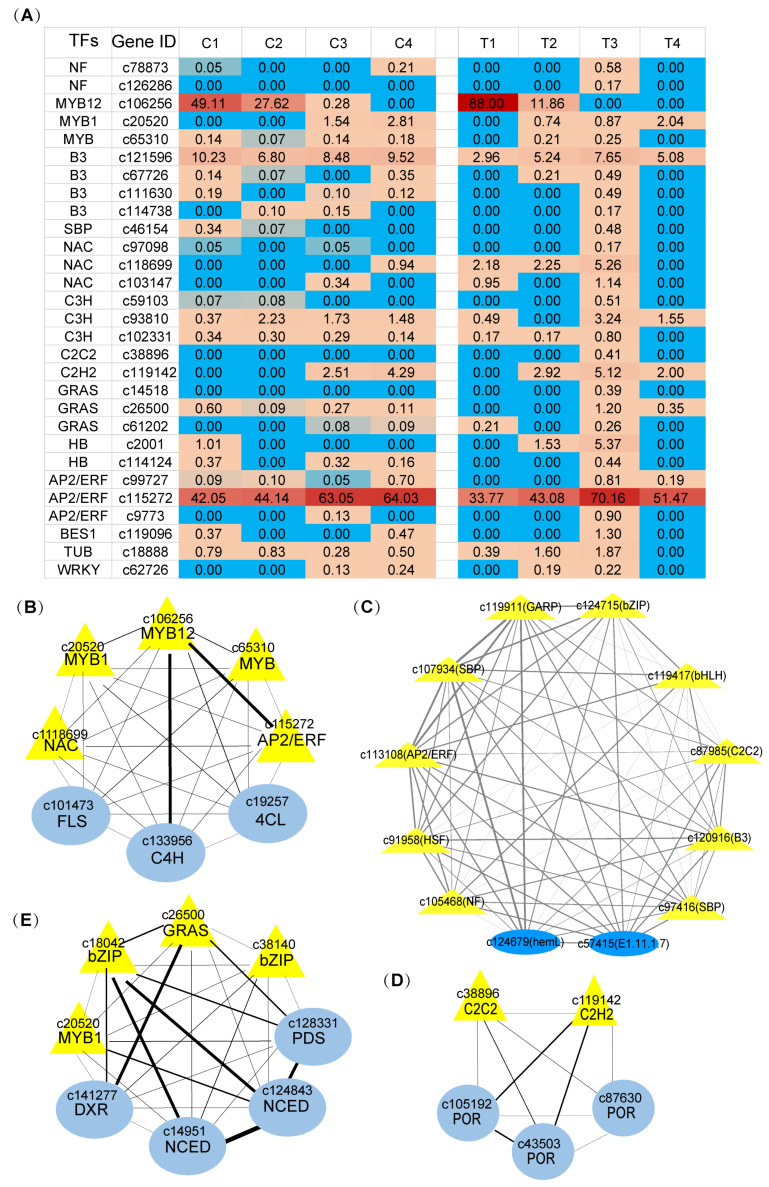
Coexpression network of transcription factor and structure genes related to floral pigment biosynthesis and the floral fragrance biosynthesis pathway during flower development. (**A**). 28 transcription factors related to pigment biosynthesis and the floral fragrance biosynthesis pathway during flower development; from blue to red indicated the FPKM values of genes from low to high, and the value is indicated in the box. (**B**–**E**). Coexpression network of transcription factors (yellow triple angle) and structure genes (blue circle) related to floral pigment. (**B**), flavonoids biosynthesis pathway; (**C**), chlorophyll biosynthesis pathway; (**D**), chlorophyll biosynthesis pathway; (**E**), carotenoid metabolism pathway and floral fragrance (**B**), benzenoid/phenylpropanoid biosynthesis pathway; (**D**), chlorogenic acid biosynthesis pathway) biosynthesis pathway. Edge width indicates the strength of the relation between two genes.

**Figure 8 ijms-22-08249-f008:**
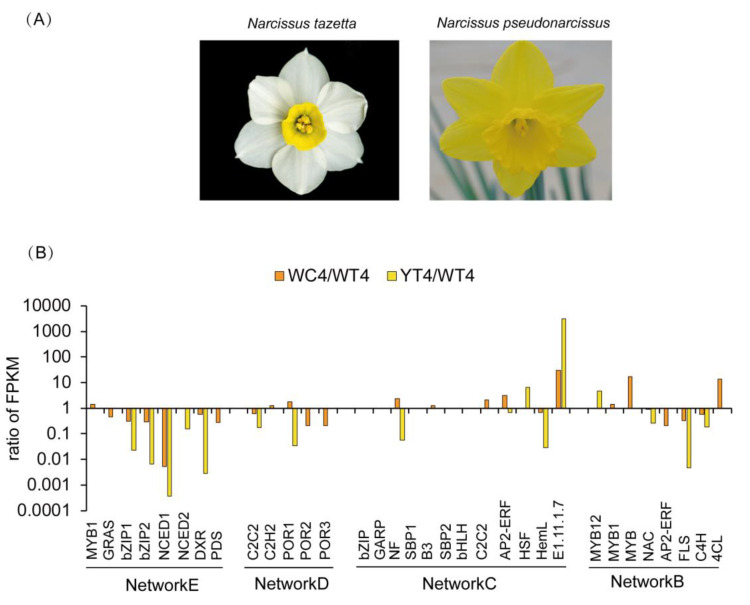
Expression pattern of candidate TFs in *Narcissus tazetta* and *Narcissus pseudonarcissus* by RNA-seq dataset. (**A**). Flowers of *Narcissus tazetta* and *Narcissus pseudonarcissus* at T4 stage. (**B**). Ratio of FPKM from the RNA-seq dataset of coronas, tepals of *Narcissus tazetta* and tepals of *Narcissus pseudonarcissus* at T4 stage. WC4, coronas of *Narcissus tazetta* at T4 stage, WT4, tepals of *Narcissus tazetta* at T4 stage; YT4, tepals of *Narcissus pseudonarcissus* at T4 stage. Network B-E refer to Figure 7B–E.

## Data Availability

GenBank Short Read Archive (Accession SRP083093). And https://submit.ncbi.nlm.nih.gov/subs/SUB10083597.

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
