# Peer review of "Transcriptome-Based WGCNA Analysis Reveals Regulated Metabolite Fluxes between Floral Color and Scent in Narcissus tazetta Flower"

_ijms, 2021, doi:10.3390/ijms22158249_

Round 1
Reviewer 1 Report
The manuscript by Yan et describes an analysis for fragrance and pigmentation of Narcissus flower. The study is interesting. The experimental design is particularly fine: tissue- specific transcript profiles (tepals and corona) during their development. But the authors could go further in their research and provide further evidence for their claim of a competition between the biosynthetic pathways that lead to pigmentation and/or fragrance. It would be interesting to present a functional analysis of selected candidate genes. Alternatively, take advantage of natural variation, and present a similar analysis among other variants/ species of Narcissus flowers (there are species that the tepals and corona are both yellow).
A major concern is that the manuscript is very dense to the reader. Sometimes, in the same section, it is described methodological details together with the results and their discussion. It is quite tangling. Also, and probably consequently, sometimes, it is not clear what are the actual results and previous research. The advice is to separate the results and discussion and to present a shorter conclusion section.
Reviewer 2 Report
This study clarified the possible molecular mechanisms of floral color and fragrant formation based on metabolic and transcriptomic data. It provides a basis for further floral pigment and floral fragrance in plants. Overall, the quality of this study is good but there are some points need to be fixed before considering publication.
- Figure 2 add the legend bar text on C and E panels.
- Figure 2 it is hard to see Carotenoids decline in D panel.
- line 162 Are there two T1 samples?
- line 167 How to obtain the DEGs?
- line 175 is there a way to use adjust p value to be the cutoff?
- Figure 4 it is better to use the figure instead of table to show the expression patterns. Also, more information such as what’s meaning of each row of the small table needs to be clarified.
- Figure 7 B. the thickness of each line connecting between two TFs needs to be added. In general, this could indicate relations are strong or not.
- Authors address some important relationships among different TFs. How do we know their relations are true or not? Could we find a way to validate sort of them?
- In methods, lots of information for how to process transcriptomic data are missing like data QC (quality control), mapping tools, etc.
